# Transient hysteresis and inherent stochasticity in gene regulatory networks

M. Pájaro [1,3], I. Otero-Muras [1,3], C. Vázquez [2] & A.A. Alonso [1*]

Cell fate determination, the process through which cells commit to differentiated states is commonly mediated by gene regulatory motifs with mutually exclusive expression states. The classical deterministic picture for cell fate determination includes bistability and hysteresis, which enables the persistence of the acquired cellular state after withdrawal of the stimulus, ensuring a robust cellular response. However, stochasticity inherent to gene expression dynamics is not compatible with hysteresis, since the stationary solution of the governing Chemical Master Equation does not depend on the initial conditions. We provide a quantitative description of a transient hysteresis phenomenon reconciling experimental evidence of hysteretic behaviour in gene regulatory networks with inherent stochasticity: under sufficiently slow dynamics hysteresis is transient. We quantify this with an estimate of the convergence rate to the equilibrium and introduce a natural landscape capturing system's evolution that, unlike traditional cell fate potential landscapes, is compatible with coexistence at the microscopic level.

[1] BioProcess Engineering Group, IIM-CSIC. Spanish National Research Council, Eduardo Cabello 6, 36208 Vigo, Spain. [2] Department of Mathematics, University of A Coruña, Campus Elviña s/n, 15071 A Coruña, Spain. [3]These authors contributed equally: M. Pájaro, I. Otero-Muras. *email: antonio@iim.csic.es

In a deterministic description, binary decision making is attributed to the irreversible state transition between two mutually exclusive stable steady states in response to a signal. This state transition is usually governed by regulatory motifs with the capacity for bistability and hysteresis[1], thus ensuring that the system does not switch back immediately when the signal is removed[2].

The stochastic dynamic behavior of a gene regulatory network is governed by a chemical master equation (CME), which describes the time evolution of the probability distribution of the system state. The stationary solution of the CME is unique and independent on the initial state of the system[3] and therefore, incompatible with memory effects or hysteresis. The incompatibility of hysteresis with intrinsic noise in gene regulatory networks has been addressed, for example, by Lestas et al.[4]. However, there are numerous works providing experimental evidence of hysteretic behavior under significant levels of stochasticity[5–8].

In the context of phenotypic switching and cell fate determination, three different scenarios have been distinguished and experimentally observed for binary decision making: deterministic irreversible[9–11], stochastic reversible[12], and stochastic yet irreversible state transitioning[13]. Reversibility is understood here as the capacity of individual cells to switch back in absence of external signals. According to a pseudo-potential interpretation, dynamics are directed by a pseudo-potential landscape divided by a separatrix into two basins of attraction such that each local minimum corresponds to a specific cellular state. Stochastic irreversible transitions are found to appear when cells are initialized on (or near) the separatrix[13].

In this article we provide a quantitative description of hysteresis and apparent irreversibility in stochastic gene regulatory networks at the single cell level as transient effects, which disappear at the stationary state. Our analysis is based on an accurate approximation of the CME. This means that our results are valid for purely stochastic regimes far from the thermodynamic limit, and thus complementary to those based on the classical linear noise approximation for systems closer to the thermodynamic limit[4,14]. Since the stationary solution of the CME is unique[3], if the solution corresponds to a bimodal distribution, state transitions at the single cells level occur necessarily in a random and spontaneous manner, switching back and forth between regions of high probability.

Fang et al.[15] experimentally determined an energy potential-like landscape as the negative logarithm of the probability distribution, as well as the transition rates, based on previous theoretical studies[16]. In this contribution, we provide a theoretical basis that explains coexistence of different expression states. In fact, under the assumption of protein bursting[17], we propose an efficient form of the CME[17,18] that allows us to construct a meaningful probability based landscape. Furthermore, a clear link between the characteristic kinetic parameters of regulation dynamics and the resulting landscape is established.

## Results

**Deterministic description.** We consider the simplest gene regulatory motif exhibiting hysteresis, a single gene with positive self-regulation (see Supplementary Fig. 1). In its deterministic description, the evolution of the amounts of mRNA and protein X ($m$ and $x$, respectively) for the self-regulatory gene network is given by the set of ODEs:

$$\frac{\mathrm{d}m}{\mathrm{d}t} = k_m c(x) - \gamma_m m \qquad (1)$$

$$\frac{\mathrm{d}x}{\mathrm{d}t} = k_x m - \gamma_x x, \qquad (2)$$

where $\gamma_m$ and $\gamma_x$ are the mRNA and protein degradation rates, respectively. $k_m c(x)$ is the transcription rate, that is essentially proportional to the input function $c(x)$ which collects the expression from the activated and inactivated promoter states. This function incorporates the effect of protein self-regulation and takes the form[19,20]:

$$c(x) = (1 - \rho(x)) + \rho(x)\varepsilon, \qquad (3)$$

with $\rho(x)$ being a Hill function[21] that describes the ratio of promoter in the inactive form as a function of bound protein:

$$\rho(x) = \frac{x^H}{x^H + K^H}. \qquad (4)$$

The above expression, can be interpreted as the probability of the promoter being in its inactive state, where $K = k_{\mathrm{off}}/k_{\mathrm{on}}$ is the equilibrium binding constant and $H \in \mathbb{Z} \setminus \{0\}$ is an integer (Hill coefficient) which indicates whether protein X inhibits ($H > 0$) or activates ($H < 0$) expression. Finally, expression (3) includes basal transcription or leakage with a constant rate $\varepsilon = k_\varepsilon / k_m$ (see Supplementary Fig. 1) typically much smaller than 1. The parameters of the Hill function employed along the paper are $H = -7$ (the value taken from To and Maheshri[22]) and $K = 100$, whereas $\varepsilon = 0.05$. Unless other value is indicated, we use $a = 54$. Assuming that $mRNA$ degrades faster than protein X we have that $m^* = k_m c(x)/\gamma_m$ and model (1) reduces to:

$$\frac{\mathrm{d}x}{\mathrm{d}\tau} = -x + abc(x), \qquad (5)$$

where $\tau = t\gamma_x$, $a = k_m/\gamma_x$, and $b = k_x/\gamma_m$. Along the paper we use the values $\gamma_x = 4 \times 10^{-4}$ s$^{-1}$ and $\gamma_m = 20\gamma_x$ s$^{-1}$, taken from Friedman et al.[23].

The self-regulatory network described by the deterministic Eqs. (1) and (2) shows bistability and hysteresis (see Fig. 1a). For a range of the control parameter $b$ the system evolves toward one stable state or another depending on the initial conditions. We therefore say that the system has memory, since steady state values provide information about the system's past. In systems with hysteresis (dependency of the state of the system on its past), forward and reverse induction experiments follow different paths resulting in a hysteresis loop (the system switches back and forth for different values of the control parameter)[24].

**Stochastic description.** Gene expression is inherently stochastic. Taking into account that mRNA degrades faster than protein X in most prokaryotic and eukaryotic organisms[25], protein is assumed to be produced in bursts[19,20,23,26] at a frequency $a = k_m/\gamma_x$ (see Eq. (5)). From this assumption, it follows[20,23] that the temporal evolution of the associated probability density function $p : \mathbb{R}_+ \times \mathbb{R}_+ \to \mathbb{R}_+$ can be described by a partial integro-differential equation (PIDE) of the form:

$$\frac{\partial p(\tau, x)}{\partial \tau} - \frac{\partial [x p(\tau, x)]}{\partial x} = a \int_0^x \omega(x - y) c(y) p(\tau, y) \, \mathrm{d}y - ac(x) p(\tau, x), \quad (6)$$

where $x$ and $\tau$ correspond with the amount of protein and dimensionless time, respectively. The latter variable is associated to the time scale of the protein degradation, as in the previous deterministic description. In addition, $\omega(x - y)$ is the conditional probability for protein level to jump from a state $y$ to a state $x$ after a burst, which is proportional to

$$\omega(x - y) = \frac{1}{b} \exp\left[ \frac{-(x - y)}{b} \right], \qquad (7)$$

with $b$, as in Eq. (5), representing the burst size. The stationary form of the one dimensional Eq. (6) has analytical solution[19,20] $p^*(x) = C[\rho(x)]^{\frac{a(1-\varepsilon)}{H}} x^{-(1-a\varepsilon)} e^{-\frac{x}{b}}$ where $\rho(x)$ is defined in (4) and $C$

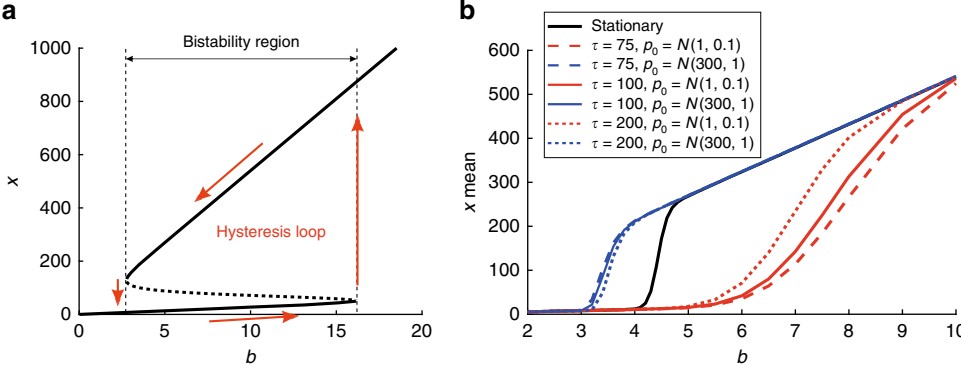

**Fig. 1** Hysteresis in deterministic vs stochastic descriptions. **a** Hysteresis loop of the deterministic self-regulatory system (positive roots of Eq. (5)). For values of the control parameter $b$ below a given threshold, there is a unique stable steady state of low protein $x$ toward which the system evolves independently of the initial conditions. For input signals above a second threshold, the system evolves toward a unique stable steady state of high $x$. For signal values within both thresholds, the system is bistable, and evolves toward one stable state or another depending on the initial conditions. In the bistability region, enclosed by two saddle-node bifurcations, three different steady states coexist for a given $b$ (stable and unstable branches are depicted using solid and dotted lines, respectively). **b** Transient hysteresis in the stochastic self-regulatory system: slow transients lead to multiple mean states leading to a transitory hysteretic behavior. Red and blue lines are transient solutions obtained from two different initial conditions in the form of Gaussian distributions $\mathcal{N}(\mu, \sigma)$ with mean $\mu$ and standard deviation $\sigma$. When the system achieves the stationary state (black solid line corresponds to the stationary solution of the PIDE model), there is a unique mean $x$-value for given $b$ (hysteresis disappears). As time increases, the solution gets closer to the stationary distribution. Simulations have been carried out in SELANSI[18]

is a normalizing constant such that $\int_0^\infty p^*(x) = 1$. It has been shown that the equilibrium solution associated to a CME is unique and stable[3]. This is also the case for the Friedman Eq. (6) whose stability has been recently proved by entropy methods[27,28], which eventually makes it to qualify as a master equation itself. It is important to remark that stability properties remain valid for higher dimensions (i.e., multiple genes and proteins). While the mean $x$-values of the stationary solution do not depend on the initial conditions, the means obtained at the transients depend on the initial number of proteins (Fig. 1b).

Note that under sufficiently slow dynamics, transient values may look stationary, thus leading to plots (red and blue lines) that resemble hysteresis, as different mean values coexist within a given interval of the $b$ parameter. Interestingly, this interval coincides with bimodal distributions in which the two most probable states are separated by a region, in the protein space, with very low probability. This explains recent experimental observations[29] in which the range of apparent hysteresis was found to shrink with time. Here we denote this phenomenon as transient hysteresis and show how, in fact, the low probability region acts as a barrier that hinders transitions between low and high protein expression, contributing in this way to slow down the dynamics toward the corresponding stationary distribution. Supplementary Fig. 2 compares transient and stationary distributions for different values of the control parameter and different initial conditions. This figure provides a clear illustration of how, in presence of stochasticity, hysteresis is transitory: it shrinks with time and disappears as the system achieves the stationary state.

In order to compute an estimate of the convergence rate to equilibrium we make use of entropy methods[28,27] and define the entropy norm as $G = \int_0^\infty H(u(\tau, x))p^*(x)\mathrm{d}x$ where $H(u(\tau, x))$ is a convex function in $u$, that in this study has been chosen to be $H(u) = u^2 - 1$, with $u = p(\tau, x)/p^*(x)$. According to Pájaro et al.[28] and Cañizo et al.[27], $G$ satisfies the following differential inequality:

$$\frac{\mathrm{d}G}{\mathrm{d}\tau} \leq -\eta G, \qquad (8)$$

where $\eta$ is a positive constant (its dimension is the inverse of time) related to regulation (parameters $H$ and $K$), as well as the

transcription-translation kinetics $(a, b)$. The smaller $\eta$, the slower its convergence toward the corresponding equilibrium solution. Computing $\eta$ requires a full simulation of (6) until the system reaches the equilibrium distribution for each parameter on a given range, what is computationally involved. In this work, the PIDE model (6) is solved by using the semi-lagrangian method implemented in the toolbox SELANSI[18].

Alternatively, we provide a truncation method to approximate the rate of convergence that we use here for verification purposes. The method makes use of the discrete jump process representation (see Supplementary Fig. 3), which is a precursor of Friedman PIDE model, by making the protein amount a continuous variable[17]. With this method (see Methods section) we compute the negative eigenvalue with smallest absolute value of the state change matrix $\mathcal{M}$ which we refer to as $\lambda_1$. This eigenvalue is a good approximation of the convergence rate $\eta$.

Figure 2 compares the eigenvalue $\lambda_1$ with the convergence rate $\eta$ obtained by simulation, for different values of the parameter $b$. In the parameter range where bimodal distributions occur, the negative eigenvalue $\lambda_1$ is a good approximation of the convergence rate of the PIDE model. The figure also shows how the smaller $\eta$ values correspond to the solution near equilibrium which lies within the hysteresis region in the $b$ parameter space. Remarkably, low convergence rates coincide with the parameter region in which bimodal behavior take place.

The estimation of the convergence rate (either in terms of $\eta$ or $\lambda_1$) can be obtained from kinetic coefficients $a$ and $b$ previously estimated from experiments. To that purpose, we can use the PIDE model to find by least squares from typically time dependent distributions obtained from a cell population by flow cytometry, the best set of parameters. Alternatively, distributions could be reconstructed from single cell time series. With the resulting model, simulations will be executed to estimate rate of convergence.

This example has served as a proof of concept to clarify how hysteresis, as it is known in deterministic nonlinear systems (i.e., as a long-term stationary phenomenon) has not an equivalence in a microscopic world governed by a CME. For stochastic systems, hysteretic behavior is a transitory phenomenon, i.e., it can be only obtained under transients that may resemble stationary solutions

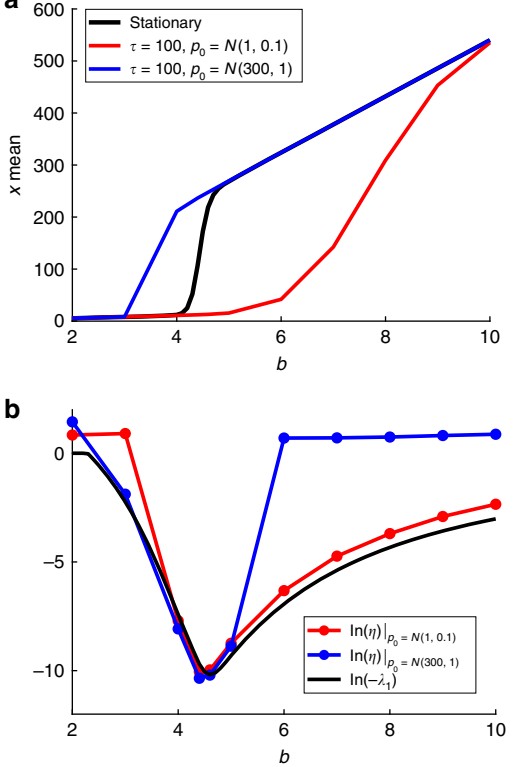

**Fig. 2** Parameter region leading to bimodal distributions. **a** Mean $x$-values plotted as a function of parameter $b$ for different initial conditions. Simulations have been carried out in SELANSI[18]. **b** Convergence rates of the solution ($\eta$ from (8) and $\lambda_1$ from matrix $\mathcal{M}$ expressed in units of inverse of time) toward the equilibrium distribution in logarithmic scale. Such slow dynamics is responsible for the phenomenon of transient hysteresis. If the system is allowed to achieve the equilibrium, hysteresis disappears. The parameter region leading to bimodal distributions corresponds with the slowest convergence rates

due to the extremely slow dynamics at which bimodal distributions evolve. Nonetheless, some correspondence can be drawn between the most frequently visited states on a microscopic system and the stable states on the deterministic counterpart (see Supplementary Note 1).

Figure 3 shows that the logarithm of the eigenvalue decreases as the parameters $a$ and $b$ become higher and smaller, respectively. Variations of the logarithm of the eigenvalue are more pronounced inside the bimodal (if one of the peaks lies at zero the bimodal distribution is also known as binary) and bistable regions. Moreover, as discussed by Pájaro et al.[30], as the parameter $a$ increases the system approaches the thermodynamic limit.

**Hysteresis in a mutual repression gene network**. We consider the gene regulatory network in Ellis et al.[31] and Wu et al.[13], where the LacI promoter is repressed by the protein expressed by the TetR promoter and vice versa, and ATc is used to inhibit the expression of TetR (see details in Supplementary Note 2).

We simulate the dynamics from two different initial conditions, $p_0 = \mathcal{N}([600, 10], 5I)$ and $p_0 = \mathcal{N}([50, 200], 5I)$, and take snapshots at 50, 100, and 150 h. In Fig. 4, we depict the dose–response curves at $t = 100$ h for each initial condition (red and blue lines, respectively) and the stationary dose–response curve. It can be observed clearly how hysteresis disappears at the stationary. Note that the transient hysteresis observed at $t = 100$ h is in agreement with experimental observations by Wu et al.[13].

The transient distributions are depicted in Supplementary Figs. 4 and 5 representing the corresponding marginal distribution for the same snapshots. As it is shown, the distribution at 50 h resembles a stationary distribution, since no significant differences are observed with those obtained at $t = 100$ h and even at $t = 150$ h. However, comparing those distributions with the stationary distribution (see also third row in Supplementary Fig. 6), we clearly conclude that the system is not at the stationary state. Thus, the corresponding dose–response curve at $t = 50$ h describes a transient hysteresis phenomenon. Note that as shown in Supplementary Figs. 6 and 7 even snapshots taken at much longer times (e.g., 1500 h) still differ significantly from the stationary solution.

The results for $t = 50$ h are coherent with the observation by Wu et al.[13] that if a trajectory starts clearly within one of the basins of attraction remains there for a long time. Note that the time needed to reach the stationary state might be longer than the natural timescales of relevance to the process. This is in accordance with Wu et al.[13] where the transitions are characterized as stochastic yet irreversible.

The convergence rates of the solution toward the equilibrium are depicted in Fig. 5. As it happens for the 1D example in Fig. 2 the parameter region leading to bimodal distributions corresponds with the slowest convergence rates; such slow dynamics is responsible for the phenomenon of transient hysteresis.

Supplementary Fig. 8 compares the set of stable and unstable equilibrium states obtained from a deterministic representation with the most and least probable microscopic states, respectively. Note that this equivalence does not support the existence of long-term (stationary) hysteresis at the microscopic level. Essentially, what the picture shows is that, rather than a parameter-dependent preferential state among two stable ones, there are two highly probable states that coexist for a given parameter region on a cell population.

## Discussion

These results provide us with an important insight on how to interpret experimental results showing hysteretic behavior at the level of gene regulatory networks: if the system is governed by the CME, hysteresis is necessarily transient. Note that for slow dynamics (high $a$ and low $b$ values) the time needed to reach the stationary state might be longer than the natural timescales of relevance to the process. This is in accordance with previous studies reporting large mean passage times[14] and also with Wu et al.[13] where they engineer a synthetic switch with stochastic yet irreversible transitions (the same mutually inhibitory gene regulatory motif is analyzed next using our PIDE approach).

The characterization of a cell response as hysteretic or non-hysteretic is important. For example, in a a recent study concerning epithelial to mesenchymal transition (EMT), a process through which epithelial cells transdifferentiate into a mesenchymal cell fate, the authors characterize two types of responses, hysteretic and non-hysteretic EMT, and report the notable influence of hysteresis on the metastatic ability of cancer cells[32].

Invoking pseudo-potential concepts to interpret dynamics in GRN under fluctuations[13], although attractive from an intuitive point of view, may be misleading since it cannot capture the notion of coexistence. By coexistence we mean that two different protein expression levels coinciding with the peaks of the bimodal distribution coexist on a cell population (assuming no cell to cell variability on the initial conditions).

The pseudo-potential landscape is not easy to compute either, specially when increasing the number of proteins expressed. Alternatively, we can use the stationary solution of (6) to

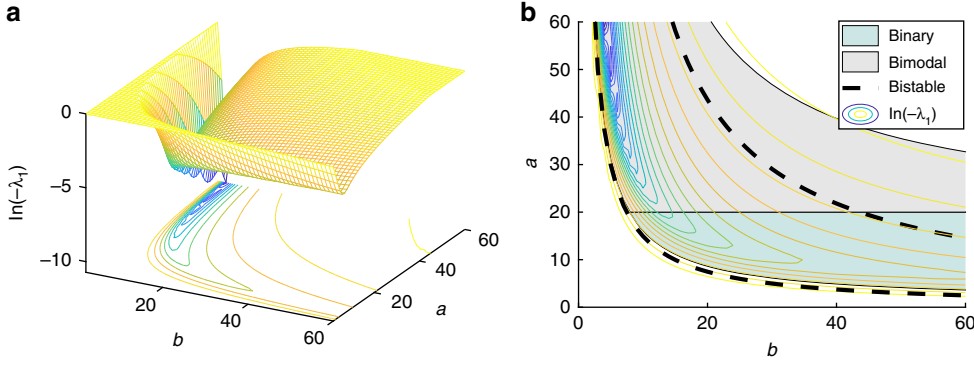

**Fig. 3** Evolution of the eigenvalue and bimodality and bistability regions in the parameter space. **a** Eigenvalue $\lambda_1$ in the parameter space computed from matrix $\mathcal{M}$ (logarithmic scale). **b** Contour of $\lambda_1$ in the parameter space. Regions of bimodality and bistability are computed by the algorithm in Pájaro et al.[20] (logarithmic scale). The figure shows how the eigenvalue evolves with parameters $a, b$

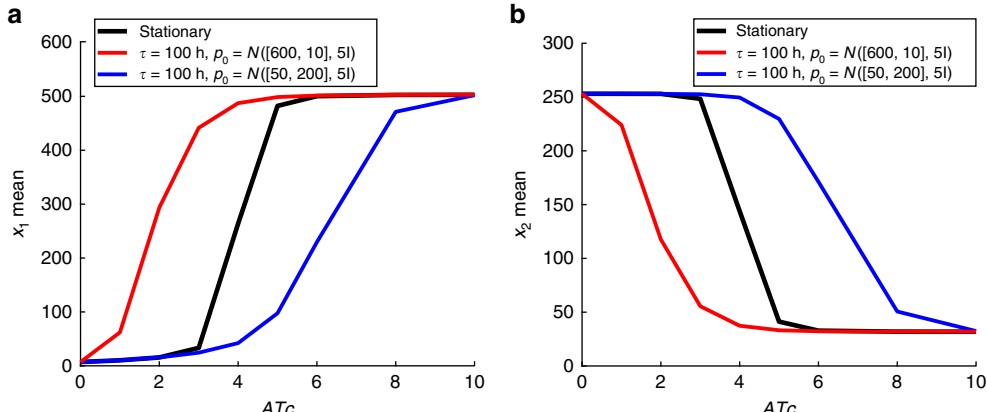

**Fig. 4** Transient hysteresis in the mutual repression gene network. **a** Mean states depend on initial conditions showing transitory hysteretic behavior for LacI. **b** Mean states depend on initial conditions showing transitory hysteretic behavior for TetR. Red and blue lines are transient solutions ($t = 100$ h) obtained from two different initial conditions in the form of multivariate Gaussian distributions $\mathcal{N}(\boldsymbol{\mu}, \boldsymbol{\Sigma})$ with mean vector $\boldsymbol{\mu}$ and covariance matrix $\boldsymbol{\Sigma}$. When the system achieves the stationary state (black solid line corresponds to the stationary solution of the PIDE model), there is a unique mean $x$-value for given $ATc$ (hysteresis disappears). Initial conditions were chosen to be near the peaks of the stationary distribution. Simulations have been carried out in SELANSI[18]

construct on the natural framework of probability distributions, a landscape informing of the possible transitions or evolution of the underlying microscopic system. As we illustrate in the example discussed in the supplementary material, its computation can be extended in a straightforward manner to larger dimensional protein spaces. This can be of use to efficiently identify most prevalent phenotypes coexisting on a given cell population.

The main assumption of the PIDE model is protein bursting (mRNA degrading faster than proteins). As reported in Pájaro et al.[17], the approximation remains generally valid even for degradation rate ratios around 2–3 (5 in the most restrictive cases). In terms of protein copy numbers, although the PIDE model is valid in any range, we expect a significant effect of the inherent intrinsic noise for low copy numbers (in the order of thousands and lower). Note that, for prokaryotic cells this is the case for the majority of the proteins[33]. Although in eukaryotic cells proteins are in general more abundant there is still a significant portion of the cases for which the copy numbers appear to be low (see for example, Schwanhausser et al.[34], Shi et al.[35], Nguyen et al.[36]). We would like to remark that extrinsic noise is not taken into account in this study since we are quantifying the effect of intrinsic noise in hysteresis.

## Methods

**Stochastic model and simulation.** We use the PIDE model[17] described in Eq. (6). The model is simulated by a semi-lagrangian method implemented in the toolbox SELANSI[18].

**Rates of convergence (truncation method).** Let $\mathcal{P} : \mathbb{R}_+ \times \mathbb{N} \to [0, 1]$, be the probability of having $n$ proteins at time $\tau = \gamma_x t$. The time evolution of $\mathcal{P}(\tau, n)$ is given by the following CME with jumps that reads

$$\frac{d\mathcal{P}(\tau, n)}{d\tau} = \sum_{i=0}^{n-1} g_i^n \mathcal{P}(\tau, i) - \sum_{i=n+1}^{\infty} g_n^i \mathcal{P}(\tau, n) + (n+1)\mathcal{P}(\tau, n+1) - n\mathcal{P}(\tau, n),$$

(9)

where the transition probability $g_i^j$ is proportional to the production rate of messenger RNA, so that

$$g_i^j := \frac{a}{b} c(i) e^{\frac{i-j}{b}}, \quad \forall j > i. \tag{10}$$

In order to obtain an approximation of the convergence rate of the PIDE model toward the stationary state, we use the truncated form of the discrete Eq. (9). Let $N$ be the maximum possible number of proteins. Then, Eq. (9) can be written in matrix form as

$$\frac{d\mathcal{P}(\tau, n)}{d\tau} = \mathcal{M}\mathcal{P}(\tau, n), \tag{11}$$

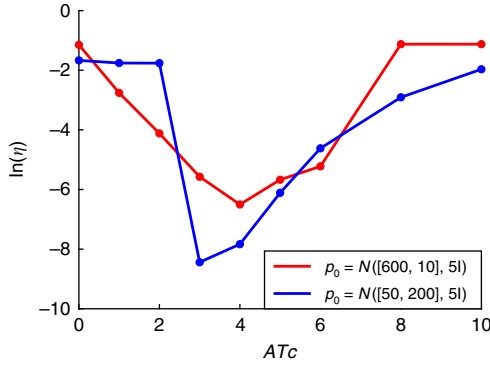

**Fig. 5** Convergence rates toward the equilibrium distribution. $\eta$ in Eq. (8) in units of $h^{-1}$ is plotted for different initial conditions (in logarithmic scale). The parameter region leading to bimodal distributions corresponds with the slowest convergence rates

where the matrix $\mathcal{M}$ reads

$$\mathcal{M} = \begin{pmatrix} -d_0 & 1 & 0 & \cdots & 0 & 0 & 0 \\ g_0^1 & -d_1 & 2 & \cdots & 0 & 0 & 0 \\ g_0^2 & g_1^2 & -d_2 & \ddots & 0 & 0 & 0 \\ \vdots & \vdots & & \ddots & \ddots & & \vdots \\ g_0^{N-2} & g_1^{N-2} & g_2^{N-2} & \cdots & -d_{N-2} & (N-1) & 0 \\ g_0^{N-1} & g_1^{N-1} & g_2^{N-1} & \cdots & g_{N-2}^{N-1} & -d_{N-1} & N \\ g_0^N & g_1^N & g_2^N & \cdots & g_{N-2}^N & g_{N-1}^N & -d_N \end{pmatrix}, \qquad (12)$$

with the elements of the diagonal $d_i$ being of the form

$$d_i = \begin{cases} lli + \sum_{n=i+1}^{N} g_i^n & \text{if } i = 0, \dots, N-1, \\ N & \text{if } i = N, \end{cases} \qquad (13)$$

equivalently

$$d_i = i + \frac{ac(i)}{b\left(e^{\frac{1}{b}}-1\right)}\left(1 - e^{\frac{i-N}{b}}\right) \text{ for } i = 0, \dots, N. \qquad (14)$$

The steady state is given by the null space of matrix $\mathcal{M}$, which is spanned by the normalized eigenvector associated to the unique zero eigenvalue, as the associated eigenspace has dimension one. Actually, since the graph associated to matrix $\mathcal{M}$ (See Supplementary Fig. 3) has one trap, all the eigenvalues are negative except one (which is zero)[37]. By $\lambda_1$, we denote the negative eigenvalue closer to zero, i.e., the one with smallest absolute value.

**Reporting summary**. Further information on research design is available in the Nature Research Reporting Summary linked to this article.

## Data availability
All relevant data needed to reproduce the results are included in the text and supplementary information.

## Code availability
The semi-lagrangian method to simulate the PIDE model is freely avaliable and can be downloaded at: https://github.com/selansi/Selansi.

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

## Acknowledgements
M.P. and A.A.A. acknowledge funding from grant PIE201870E041; I.O.M. acknowledges funding from Spanish MINECO (and the European Regional Development Fund) project SYNBIOCONTROL (grant number DPI2017-82896-C2-2-R). C.V. has been partially funded by the spanish MINECO project MTM2016-76497-R and Xunta de Galicia grant ED431C2018/033.

## Author contributions
A.A.A. and I.O.M. conceived the research. M.P. and A.A.A. performed the research. C.V., M.P., I.O.M., and A.A.A. contributed to the simulation methods. A.A.A., M.P., I.O.M., and C.V. wrote the paper. A.A.A. supervised the project.

## Competing interests
The authors declare no competing interests.
