## [Peer Review File · Nature Communications]

Reviewers' comments:

Reviewer #1 (Remarks to the Author):

The paper by Pajaro et al reports on a stochastic study whose result claims that noise in gene regulatory systems precludes hysteresis. While the topic is certainly of wide interest to the community of modelers in biology, it is my opinion that the study is not performed rigorously enough to make the purported claims. Also there is a recent paper which has related results on a very similar gene regulatory network. Given the large re-design of the study needed, I recommend rejection of the present manuscript.

My detailed comments are below:

1. The authors study an effective model given by Eq. 6. These equation is derived from the CME based on several implicit assumptions: (i) the protein is very abundant such that it can be modeled in a continuous manner (ii) gene switching is assumed to be very fast (iii) mRNA is assumed to decay much quicker than protein (iv) the system of cells all have the same parameter values, i.e. an identical system of cells. Approximation (i) is generally quite good since proteins are much more abundant usually than mRNA and promoter states. Approximation (ii) is certainly not true generally or even commonly true. For example in the paper: Kaern, Mads, et al. "Stochasticity in gene expression: from theories to phenotypes." *Nature Reviews Genetics* 6.6 (2005): 451, it is mentioned that slow promoter transitions are likely for eukaryotic cells. Also bimodality or stochastic bistability is strongly manifested in cases of slow promoter switching as for example shown in: Thomas, Philipp, Nikola Popović, and Ramon Grima. "Phenotypic switching in gene regulatory networks." *Proceedings of the National Academy of Sciences* (2014): 201400049. Approximation (iii) is valid for bacteria but it is highly doubtful for eukaryotic cells. See for example Fig. 5 of the paper Schwanhäusser, Björn, et al. "Global quantification of mammalian gene expression control." *Nature* 473.7347 (2011): 337. Approximation (iv) is perhaps the worst approximation of the four made since there is no such thing as a population of perfectly identical cells. Rate constants will vary from cell to cell and hence the means calculated from experiments over a cell population CANNOT be compared to those predicted from the current theory. Due to the unrealistic assumptions made, the theory presented cannot hence be used to make the claim that experiments which indicate hysteresis might be wrong! One would need to re-design the study to include a higher degree of realism -- in particular use the CME rather than Eq. 6 so that all assumptions can be overcome. Note that the exact steady-state solutions of the CME for positive (and negative) autoregulatory loops have been derived and one could make use of these. See the two papers:

Grima, Ramon, Deena R. Schmidt, and Timothy J. Newman. "Steady-state fluctuations of a genetic feedback loop: An exact solution." *The Journal of chemical physics* 137.3 (2012): 035104.

Kumar, Niraj, Thierry Platini, and Rahul V. Kulkarni. "Exact distributions for stochastic gene expression models with bursting and feedback." *Physical review letters* 113.26 (2014): 268105.

The time-development of these systems can be easily studied using Finite State Projection or the SSA.

2. The authors present only a study for a very simple positive regulatory feedback motif and then make a grand claim in the title and paper that their results apply to gene regulatory networks. To make such a claim they would need to study a set of commonly occurring systems where hysteresis is possible and not just one.

3. A recent paper in *Physical Review E* studies essentially their same system but makes a much more rigorous analysis of the phenomenon:

Lee, JaeJun, and Julian Lee. "Quantitative analysis of a transient dynamics of a gene regulatory network." *Physical Review E* 98.6 (2018): 062404.

Reviewer #2 (Remarks to the Author):

Pajaro et al calculate convergence rates to the equilibrium in bistable systems and suggest their use for potentials. The authors have faced a problem plaguing this field. However, both the problem and the solution presented by the authors suffer from the same problem, the ignoring of a considerable number of prior studies.

The authors are correct to state that a number of studies have ignored the fact that hysteresis shrinks with time, which has been shown both experimentally and theoretically. These studies assume that the system is deterministic despite the obvious stochastic nature. Similarly, the authors are correct to state that invoking pseudo-potential concepts to interpret the dynamics of bistable systems in the presence of fluctuations is often misleading even though they are attractive because they are visualized intuitively.

At the same time, the authors ignore a number of studies that have already addressed these issues and show that stochasticity precludes bistability and they calculate transition / convergence rates to the equilibrium. In this way, it is difficult to assess the novelty and robustness of the results the authors present. Without aiming at a comprehensive review, I suggest the following directions to extend the manuscript.

Major points:

1. The title "Inherent stochasticity precludes hysteresis in gene regulatory networks" conveys the idea that the authors discover that their main finding is the stochasticity precludes the hysteresis. However, it has been known that in the presence of stochasticity there are transitions between the two states and that hysteresis (pseudo steady-states) varies with time. Thus, the title does not indicate novelty. There are also multiple other studies that show that stochasticity converts a deterministic bistable system into a monostable one (compare Figure 3 with figure 5 in "Noise in Gene Regulatory Networks" by Vinnicombe and colleagues).

2. Only at the end of the introduction, it becomes clear that the authors aim at showing calculations of convergence rates to the equilibrium. Important overlaps with prior studies can be found here, as well.

For example, the Fig 2 in "Deterministic characterization of stochastic genetic circuits"(PNAS) shows transition rates similar to the authors' main results in Figure 4. What is the main difference between the convergence rate and the mean passage time? Is the calculation of the convergence rate more accurate? Is it more easily generalizable? How can it be measured experimentally? What are the advantages of each of these approaches?

3. The authors should also extend their study to include more complicated bistable gene regulatory networks to show that their method is generalizable.

4. Are the calculations and approximations accurate? This can be assessed by comparison to stochastic simulation of the underlying master equation.

5. The authors use an equation with an unrealistically large Hill-number ($n=7$). The authors should include gene regulatory networks with realistic Hill numbers. Some of the studies the authors cite contain experimentally determined parameters.

Minor points:

1. In general, the text is very difficult to read.

2. The authors repeatedly use terms that are defined only in their previous publication. For example, the terms binary and bimodal in Figure 4 are defined in the authors' previous publication. These can be explained in 2-3 sentences in the current manuscript.

3. Similarly, the authors make rather limited efforts to introduce the notion potential. What are they used for? What is the notion of "coexistence"?

4. The study cited to indicate that hysteresis shrinks with time is shown in "Contribution of Bistability and Noise to Cell Fate Transitions Determined by Feedback Opening" J. Mol. Biol. and the hysteresis at a single time point is shown in the cited study ([7], Cell Reports)

Rebuttal Letter: Point by point responses to comments for the article entitled “**Inherent stochasticity precludes hysteresis in gene regulatory networks**”, by Manuel Pájaro, Irene Otero-Muras, Carlos Vázquez and Antonio A. Alonso.

Reviewer #1 (Remarks to the Author):

General Comments: *The paper by Pajaro et al reports on a stochastic study whose result claims that noise in gene regulatory systems precludes hysteresis. While the topic is certainly of wide interest to the community of modelers in biology, it is my opinion that the study is not performed rigorously enough to make the purported claims. Also there is a recent paper which has related results on a very similar gene regulatory network. Given the large re-design of the study needed, I recommend rejection of the present manuscript.*

We cannot agree with the above assertions used by the reviewer to justify the rejection of our paper. A detailed justification is given below in our point by point responses, but in general terms:

- The reviewer raises a concern about the rigorousness of our approach, using wrong assertions about the assumptions and accuracy of our method: what the reviewer is basically questioning is the validity of PIDE models, which have been introduced already by Friedman. We use the same assumptions than Friedman, and the accuracy of the PIDE models has been already proven elsewhere.
- The reviewer argues that our claim might be wrong due to lack of rigorousness of the approach. The claim cannot be wrong as it is a direct consequence of the nature of the Chemical Master Equation. **Our claim does not rely on any approximation.**
- The reviewer says that there is a recent paper with a similar study. After careful reading, the only thing in common with the reference mentioned by the reviewer is that it presents a similar network as a case study. It can be easily checked that the goals and analysis are completely different: in fact, neither hysteresis nor the dependency of the transient solutions on the initial state of the cells nor the rates of convergence to bimodal distributions are even mentioned at all (see answer to comment 3 below).

Detailed Comments:

1. *The authors study an effective model given by Eq. 6. These equation is derived from the CME based on several implicit assumptions: (i) the protein is very abundant such that it can be modeled in a continuous manner (ii) gene switching is assumed to be very fast (iii) mRNA is assumed to decay much quicker than protein (iv) the system of cells all have the same parameter values, i.e. an identical system of cells.*

The reviewer is questioning the assumptions of the PIDE model (same assumptions made by Friedman (Friedman et al, 2006, Physical Review Letters, 97(16) 168302(4)).

The reviewer asserts that the assumptions of the PIDE model are unrealistic (while these are realistic and widely used). Some of the assumptions that the reviewer mentions are not even used. The reviewer manifests doubts also on the accuracy of PIDE models and refers to alternative approaches which are actually less accurate than PIDE models and more restrictive to systems close to the deterministic limit.

1.1 Approximation (i) is generally quite good since proteins are much more abundant usually than mRNA and promoter states.

To be precise, the original paper (Friedman et al, 2006, Physical Review Letters, 97(16) 168302(4)) assumes that the lifetime of mRNA is short compared to the lifetime of the protein what in turn implies that protein is produced in bursts. Naturally, this also leads to proteins being abundant compared to mRNA, although their magnitude (in the order of 10^2 - 10^3) is still far away from the thermodynamic limit in which the linear noise approximation applies. It is important to remark here that, by comparing the solution of the Friedman PIDE with SSA simulations we demonstrated that the continuous approximation implicit in Friedman is accurate for proteins numbers in any order of magnitude (Pájaro et al, Journal Theoretical Biology, 2017, 421:51-70). Moreover, as we show in the same study, the lifetimes ratios protein-to-mRNA do not need to be particularly large. Actually, the model remains valid under mRNA lifetimes five times shorter than proteins lifetimes.

1.2. Approximation (ii) is certainly not true generally or even commonly true. For example in the paper: Kaern, Mads, et al. "Stochasticity in gene expression: from theories to phenotypes." Nature Reviews Genetics 6.6 (2005): 451, it is mentioned that slow promoter transitions are likely for eukaryotic cells.

Such assumption ("gene switching must be very fast") is not needed. Actually, the original paper by Friedman and co-workers considers both cases (fast and slow transitions). Certainly, we use a Hill function, what implicitly describes fast switching, as it is the approximation typically employed in most studies on gene regulatory networks. However, our model admits any kind of input function, including those modelling slow promoter transitions as well (see Pajaro et al, 2018, Bioinformatics 34(5):893-895). See also response to Comment 1.3 below, where we compare our simulations with the ones reported in the article mentioned by the reviewer. Remarkably, the accuracy of our model (under slow promoter transition) is superior to the alternative presented there.

1.3. Also bimodality or stochastic bistability is strongly manifested in cases of slow promoter switching as for example shown in: Thomas, Philipp, Nikola Popović, and Ramon Grima. "Phenotypic switching in gene regulatory networks." Proceedings of the National Academy of Sciences (2014): 201400049.

Please note that we do not claim that bimodality is incompatible with slow promoter switching. In fact, as we show in Appendix A, our model is able to reproduce the distributions in the paper the reviewer mentions.

In that contribution, bimodality is computed as a combination of the distributions obtained for the different promoter states. Those distributions need of the linear noise approximation assumption leading to a typical stochastic differential equation (namely an equation with a drift and a Brownian process) and the corresponding Fokker-Plank equation for probability distribution. However this requires being near (or not far from) the thermodynamic limit. Such assumption can be natural for chemical system with a large number of molecules but might become questionable in regulatory networks involving a small number of proteins (in the order of hundreds). In addition, the interpretation given in the paper for expanding the original Chemical Master Equation in terms of the “size of the system” is quite questionable in the context of gene regulatory systems. Note that such parameter is needed in order to obtain the Fokker-Plank equation by means of Van Kampen expansion. Our model on the other hand is able to capture the correct distributions obtained in the mentioned paper without the need to rely on the linear noise approximation. We have included all the details of the comparison in Appendix A to this response letter. Note that the accuracy of our model (under slow promoter transition) is superior to the alternative presented there.

1.4. Approximation (iii) is valid for bacteria but it is highly doubtful for eukaryotic cells. See for example Fig. 5 of the paper Schwanhäusser, Björn, et al. “Global quantification of mammalian gene expression control.” Nature 473.7347 (2011): 337.

As we discuss in response to Comment 1.1, our proposed model remains valid for mRNA lifetimes five times shorter than proteins lifetimes. In addition, and contrary to what the reviewer asserts, the above mentioned Figure 5 shows that in many cases the lifetime of proteins is larger or much larger than the lifetimes of mRNA (data are in a logarithmic scale). Moreover, as discussed in the corrigendum to “Global quantification of mammalian gene expression control” (Nature 473, 337–342 (2011); doi:10.1038/nature10098) the correct figures for protein lifetimes should be even higher what is in support of the assumption even for eukaryotic cells. The paragraph reads as follows:

“Mark Biggin of the Lawrence Berkeley National Laboratory contacted us, noting that our mass-spectrometry-based protein copy number estimates are lower than several literature-based values. We therefore re-analysed the scripts used for data processing, and found a scaling error that occurred during the conversion of normalized protein intensity values into absolute copy number estimates.”

1.5 Approximation (iv) is perhaps the worst approximation of the four made since there is no such thing as a population of perfectly identical cells. Rate constants will vary from cell to cell and hence the means calculated from experiments over a cell population CANNOT be compared to those predicted from the current theory.

Certainly, we do not study kinetic coefficient variability within a cell population, since what we want to explore is the effect of intrinsic noise on the dynamics of bimodal distributions and eventually its role on the phenomenon of hysteresis.

On the other hand, such theoretical consideration (same kinetic coefficients) intends to keep

things as simple as possible as it is the case in most literature on analysis of gene regulatory networks, what includes the papers mentioned by the reviewer (e.g. the one in PNAS, 2014; J. Chem Physics, 2012; Physics Rev E, 2018). Our goal is to clarify the role of intrinsic noise, namely how to reconcile hysteresis with intrinsic stochasticity in gene regulatory networks. Therefore, adding kinetic parameter variations makes no sense in this context, as it will blur the output overlapping different noise sources.

1.6 Due to the unrealistic assumptions made, the theory presented cannot hence be used to make the claim that experiments which indicate hysteresis might be wrong! One would need to re-design the study to include a higher degree of realism – in particular use the CME rather than Eq. 6 so that all assumptions can be overcome.

The CME solved directly for one gene case and via SSA when more genes are involved, have been used in all our studies to verify the validity of Eqn 6. As we demonstrated in previous works (listed in the reference section) both methods showed a perfect agreement with the solution of Eqn 6, what supports its applicability. Hence, the claims we make based on Eqn 6 cannot be wrong. On the other hand, increasing realism by considering parameter variability from cell-to-cell is out of the scope and makes no sense in this study by the reasons discussed in the response to Comment 1.5. In conclusion, we do not agree with any of the arguments given by the reviewer to justify a redesign of the research.

*1.7 Note that the exact steady-state solutions of the CME for positive (and negative) autoregulatory loops have been derived and one could make use of these. See the two papers:
Grima, Ramon, Deena R. Schmidt, and Timothy J. Newman. "Steady-state fluctuations of a genetic feedback loop: An exact solution." *The Journal of chemical physics* 137.3 (2012): 035104.
Kumar, Niraj, Thierry Platini, and Rahul V. Kulkarni. "Exact distributions for stochastic gene expression models with bursting and feedback." *Physical review letters* 113.26 (2014): 268105.*

Exact steady-state solutions with positive and negative feedback have been also derived for the Friedman equation (see [8]). In fact we use it as an additional way to evaluate the accuracy of our numerical method. However, note that in order to reconcile the (apparently contradictory) observations involving hysteresis, we need dynamic solutions rather than steady-state ones to estimate the rates of convergence. In fact any CME is not compatible with (long term) hysteresis. In presence of stochasticity, hysteresis is necessarily transient and shrinks with time.

1.8 The time-development of these systems can be easily studied using Finite State Projection or the SSA.

Actually, for the one protein case, a simple version of Finite State Projection is what we use to compute the rate of convergence via the eigenvalues. As we mentioned above, for validation purposes, we make use of SSA to solve the CME and compare with the Friedman equation in all studies, not just with one gene but including several interacting regulatory networks (see papers

by Pajaro et al (2017; 2018). However we cannot agree in that FSP or SSA are computationally efficient methods when dealing with more than one gene, as the reviewer seems to suggest. That is why we rely on the Friedman equation instead, which in addition offers a neat biological interpretation.

2. *The authors present only a study for a very simple positive regulatory feedback motif and then make a grand claim in the title and paper that their results apply to gene regulatory networks. To make such a claim they would need to study a set of commonly occurring systems where hysteresis is possible and not just one.*

We tried to keep the exposition as clear as possible using a representative yet simple genetic regulatory circuit (similarly as the works the reviewer recommends do). Our claim is based on the fact that the steady-state solution of a CME is unique, therefore it must be valid for any set of gene regulatory networks. Having said that, and also to comply with the comments 3 and 5 of reviewer #2 we have analyzed a two interacting gene network from the literature, included in the SI in the revised version. Note that such case is by no means “easily studied” with SSA or FSP and need more efficient solution methods, as for instance, SELANSI [7]. In the additional case study we reproduce with our method the experimental results by [13] in a 2 gene regulatory networks using the same experimentally determined parameters. Although they have been proved elsewhere, this case study illustrates once again the robustness and validity of PIDE model.

3. *A recent paper in Physical Review E studies essentially their same system but makes a much more rigorous analysis of the phenomenon: Lee, JaeJun, and Julian Lee. “Quantitative analysis of a transient dynamics of a gene regulatory network.” Physical Review E 98.6 (2018): 062404.*

The only relation between this work and ours is the network chosen as a case study. The work mentioned compares a deterministic and a stochastic representation of a simple yet representative network (this network happens to coincide with ours). The paper, in particular, discusses whether extinction events, associated to zero steady states in deterministic models, are possible in the presence of noise. Results show that in the absence of a baseline protein production, or when noise dominates over baseline production extinction events (related to a class of bimodal distribution known as binary) are possible but very rare and always in very long time scales (i.e. a huge number of cell generation cycles). The goal, analysis and results have nothing to do with ours. Hysteresis is not even mentioned.

Reviewer #2 (Remarks to the Author):

Pajaro et al calculate convergence rates to the equilibrium in bistable systems and suggest their use for potentials. The authors have faced a problem plaguing this field. However, both the problem and the solution presented by the authors suffer from the same problem, the ignoring of a considerable number of prior studies.

The authors are correct to state that a number of studies have ignored the fact that hysteresis shrinks with time, which has been shown both experimentally and theoretically. These studies assume that the system is deterministic despite the obvious stochastic nature. Similarly, the authors are correct to state that invoking pseudo-potential concepts to interpret the dynamics of bistable systems in the presence of fluctuations is often misleading even though they are attractive because they are visualized intuitively.

At the same time, the authors ignore a number of studies that have already addressed these issues and show that stochasticity precludes bistability and they calculate transition / convergence rates to the equilibrium. In this way, it is difficult to assess the novelty and robustness of the results the authors present. Without aiming at a comprehensive review, I suggest the following directions to extend the manuscript.

We thank the reviewer for the comments, which helped us to improve the manuscript. The reviewer's main concern is related to the fact that we have omitted some previous studies dealing with related issues (we thank the reviewer for the references), and therefore, it was difficult to assess the novelty and robustness of the results with respect to previous works. We have studied in detail the references given by the reviewer. They have been included within the discussion, helping us in fact to better justify the novelty and robustness of our approach. In the new version, the contribution of the paper is also clarified more explicitly so we hope that the lack of overlap and the relevance of our paper is now clear.

Major points:

1 The title "Inherent stochasticity precludes hysteresis in gene regulatory networks" conveys the idea that the authors discover that their main finding is the stochasticity precludes the hysteresis. However, it has been known that in the presence of stochasticity there are transitions between the two states and that hysteresis (pseudo steady-states) varies with time. Thus, the title does not indicate novelty. There are also multiple other studies that show that stochasticity converts a deterministic bistable system into a monostable one (compare Figure 3 with figure 5 in "Noise in Gene Regulatory Networks" by Vinnicombe and colleagues).

We agree with the reviewer that the title might lead to confusion about the main contribution of the paper. Our intention with the previous title was to warn researchers about possible misinterpretations of their experimental results. It was not our intention to convey the idea that "we discover" that stochasticity precludes hysteresis (this is a direct consequence of the CME, classic theory of stochastic systems). To address this concern, avoiding any confusion about this point, we modified the title to "Transient hysteresis and inherent stochasticity in gene regulatory networks".

That stochasticity precludes hysteresis is a direct consequence of the nature of the CME (the solution does not depend on the initial conditions). However, this seems not to be clear in the community, as it can be concluded from numerous recent works including theoretical/com-

putational and experimental analysis of hysteresis in the context of stochastic gene regulatory networks. The present state of the art may, in our opinion, be a cause of confusion and lead to misinterpretation of experimental results and non-consistent theoretical explanations.

On the one hand, our paper clarifies this point by providing a quantitative description that explains why experimental works show hysteresis behaviour while this is in contradiction with the theory of stochastic systems. On the other hand, our work gives a consistent theoretical interpretation of the phenomenon. To both purposes, we use a novel approach, based on an efficient and accurate computation of the CME, namely a PIDE model which validity/accuracy/robustness have been demonstrated elsewhere [1, 3, 6, 7]. In the new version of the paper, we introduced the term transient hysteresis to remark that hysteresis-like behaviour under stochastic noise is necessarily a transitory phenomenon. In this way, we differentiate among “long term” or “stationary” hysteresis (referring to the classical notion of hysteresis from the deterministic world) versus “transient” hysteresis to refer to the hysteretic behaviour observed when stochasticity is present (that disappears when the system reaches the stationary distribution).

The reviewer mentions previous studies pointing in the same direction than our work (references [5] and [9]). We hope that, in the new version, the contribution of our paper is totally clear with respect to previous works. To the best of our knowledge, after extensive revision of the literature we haven’t found further studies dealing specifically with this problem.

Regarding [5] (we were not aware of this reference, where the hysteresis phenomenon is addressed in one of the sections). This contribution is acknowledged in the introduction of the new version as follows: ”The stationary solution of the CME is unique and independent on the initial state of the system [12] and therefore, incompatible with memory effects or hysteresis. The incompatibility of hysteresis with intrinsic noise in gene regulatory networks has been addressed, for example, by Lestas *et al* [5]”

We would like to remark here that the results presented in [5] rely either on a simplified version of the CME with linear transition rates or on the classical linear noise approximation (we clarify this in the new version, see response to comment 2) The latter being valid in the large volume limit, a concept quite difficult to be interpreted on a single cell or even on a cell population context. Indeed, Figure 5 looks qualitatively similar to ours. However it should be noted that pseudo-hysteresis are obtained by artificially injecting a sinusoidal input parameter. On the contrary, our approach copes with stochastic phenomena produced by reasonably small number of particles. At this scale, transient hysteresis emerge naturally due to the slow dynamics associated to some bimodal distributions.

Moreover, reference [5] provides an interpretation of the most probable states in terms of a potential which may not generally exist in gene regulatory networks (in fact, reviewer #2 admits that this kind of interpretation is often misleading). Our work provides a novel and completely consistent interpretation of the phenomenon in terms of probability distribution/convergence rates, instead.

We are grateful to reviewer #2 for this reference (we were not aware of it), as we think that the corresponding discussion and its inclusion in new version gives an added value to our paper as well. Concerning the reference [9], we justify the lack of overlap in the next response.

2. *Only at the end of the introduction, it becomes clear that the authors aim at showing calculations of convergence rates to the equilibrium. Important overlaps with prior studies can be found here, as well. For example, the Fig 2 in “Deterministic characterization of stochastic genetic circuits” (PNAS) shows transition rates similar to the authors’ main results in Figure 4. What is the main difference between the convergence rate and the mean passage time? Is the calculation of the convergence rate more accurate? Is it more easily generalizable? How can it be measured experimentally? What are the advantages of each of these approaches?*

Concerning the second reference suggested [9], there is no such overlap with our work. First, note that in [9] the authors are working on the deterministic limit while we work on the purely stochastic regime. Second, the goal of [9] is different from ours. They compute stability phase diagrams for gene regulatory systems with noise. In that aim, they propose a method based on deterministic analysis corrected with noise since, as the authors admit:

“Unfortunately, there are few analytic methods for examining the qualitative behavior of stochastic systems. Here we describe such a method that extends deterministic analysis to include leading-order corrections due to the molecular noise”.

They illustrate the method with two case studies, a bistable positive-feedback loop and a negative-feedback oscillator. They compute stability phase diagrams by analyzing the eigenvalues of a linearized deterministic system perturbed by a stochastic fluctuation.

What they depict in Figure 2 in [9] are precisely the regions of bistability/monostability shaped by signs (negative/positive) of the eigenvalues associated to a deterministic set of equations. Although hysteresis is not explicitly mentioned in [9], note that, precisely because they work on the deterministic limit, they find regimes of bistability while in our work we quantify via stochastic analysis that long term hysteresis (and therefore bistability) is precluded. Our results are in this sense complementary as we declare now in the introduction:

“In this article we provide a quantitative description of hysteresis and apparent irreversibility in stochastic gene regulatory networks at the single cell level as transient effects, which disappear at the stationary state. Our analysis is based on an accurate approximation of the CME. This means that our results are valid for purely stochastic regimes far from the thermodynamic limit, and thus complementary to those based on the classical linear noise approximation which are suitable for systems near the thermodynamic limit [5, 9].”

In our paper, we are not interested in assessing regimes of monostability and bistability, but to reliably estimate rates of convergence to equilibrium. To this aim, we directly compute the eigenvalues associated to the CME (note that these eigenvalues cannot be positive since its solution is unique and stable) for the different kinetic parameters. These eigenvalues determine the rate of convergence of the slowest mode (others will die out faster) and shows that the slowest convergence rates coincide with parameter combinations leading to bimodal distributions. This is what Figure 4 (a) depicts, while Figure 4 (b) details the possible shapes of the distribution as a function of the kinetic parameters, and the corresponding equilibria of the deterministic counterparts.

In the supplementary info of [9] we find the following assertion, which in fact supports the same conclusion that we derive via convergence rates:

“Bistability is a property exhibited by deterministic systems. In a stochastic context, bistability is sometimes assigned to an equilibrium probability distribution with two maxima, irrespective of their separation. A more practical criterion for bistability is that the two states are long-lived and that the mean escape time from one state to the other is longer than the natural timescales in the problem.”

We have included thus the reference and the following discussion in the new version:

These results provide us with an important insight on how to interpret experimental results showing hysteretic behaviour at the level of gene regulatory networks: if the system is governed by the CME, hysteresis is necessarily transient. Note that for slow dynamics (high a and low b values), the time needed to reach the stationary state might be longer than the natural timescales of relevance to the process. This is in accordance with previous studies reporting large mean passage times [9], and also with Wu et al. [13] where they engineer a synthetic switch with stochastic yet irreversible transitions (the same mutually inhibitory gene regulatory motif is analyzed in the supplementary material using our PIDE approach).

The characterization of a cell response as hysteretic or non-hysteretic is important. For example, in a recent study concerning epithelial to mesenchymal transition (EMT), a process through which epithelial cells transdifferentiate into a mesenchymal cell fate, the authors characterize two types of responses, hysteretic and non-hysteretic EMT, and report the notable influence of hysteresis on the metastatic ability of cancer cells [2].

In the supplementary info, the authors compute the escape time for the single-variable autoactivator model by an expression based on the Fokker-Plank equation. In relation to the response to comment 1.3 from reviewer #1 the assumption of a linear noise approximation leading to a typical stochastic differential equation (namely an equation with a drift and a Brownian process) and the corresponding Fokker-Plank equation for probability distribution requires being not far from the thermodynamic limit. Such assumption can be natural for chemical systems with a large number of molecules but becomes questionable in regulatory networks involving a small number of proteins (e.g in the order of hundreds). However, our PIDE model is suitable for systems that are far from the deterministic limit.

What is the main difference between the convergence rate and the mean passage time? Is the calculation of the convergence rate more accurate? Is it more easily generalizable?

The rate of convergence is defined as the rate at which a distribution approaches a stationary one, whereas the mean first passage time is the average time needed to leave a given state of the system, this being typically applied to systems driven by random fluctuations. In this way, both concepts are different, being the mean first passage time not relevant for our arguments. Having said that, there might be some connections. In our experience, the larger the mean time to first cross from one peak of the distribution to the other, the lower the rate of convergence. At this point, our group is studying such relationship on the system described in Figure S3 of the supplementary material. We plan this to be the subject of a future work.

As indicated below, our convergence rates rely on an accurate approximation of the CME valid for purely stochastic regimes far from the thermodynamic limit. Accuracy of the convergence rate is validated on the one gene example by comparing the eigenvalues obtained from the

discrete CME with the estimation of μ in equation (8). Such inequality would be the way to generalize calculations to more complex networks.

How can it be measured experimentally? What are the advantages of each of these approaches?

Although it is difficult to directly measure rates of convergence, they can be obtained from the kinetic coefficients (a and b and degradation rates) estimated from experiments. A potential method to estimate the coefficients would be to calibrate the PIDE model, namely to estimate kinetic coefficients by least squares from experimental distributions (typically time dependent) obtained from a cell population by flow cytometry. Alternatively, distributions could be reconstructed from single cell time series. With the resulting model, simulations can be executed to estimate η in equation (8) which as we show in Figure 3 coincides with the highest (negative) eigenvalue, thus determining the rate of convergence. A paragraph has been included in the new version to discuss this question.

3. The authors should also extend their study to include more complicated bistable gene regulatory networks to show that their method is generalizable.

We tried to keep the exposition as clear as possible using a representative yet simple genetic regulatory circuit. This, in our opinion, does not risk generality, as the conclusions we reach do not depend on the complexity of the network. Nonetheless, in order to comply with the reviewer's suggestion, a two gene regulatory network is analyzed in the revised version, leading us to similar conclusions. The analysis has been included in the supplementary information. In the additional case study we reproduce with our method the experimental results by [13] in a 2 gene regulatory networks using the same experimentally determined parameters. Although they have been proved elsewhere, this case study illustrates once again the robustness and validity of PIDE model.

4. Are the calculations and approximations accurate? This can be assessed by comparison to stochastic simulation of the underlying master equation.

Yes, they are. Such comparison has been made even for cases involving more than one gene networks, what demonstrates the reliability of the so-called Friedman equation.

The CME solved directly for one gene case, or via SSA when more genes are involved, have been used in all our studies to verify the validity of Eqn 6. As we demonstrated in previous works, both methods showed a perfect agreement with the solution of Eqn 6, what supports its applicability. References to such works are included in the bibliography.

5. The authors use an equation with an unrealistically large Hill-number ($n=7$). The authors should include gene regulatory networks with realistic Hill numbers. Some of the studies the authors cite contain experimentally determined parameters.

The illustrative example has been inspired in the work by To and Maheshri [11]. This work has been cited in the revised version. The authors use a tet-Off system adapted from budding yeast, using previously designed seven binding sites promoter. In agreement with reviewer's comment, we include in the revised version (Supplementary Information) the example by Wu et al [13] with experimentally determined parameters, what includes a low Hill coefficient.

Minor points:

1. *In general, the text is very difficult to read.*

In the revised version we follow the indications of the reviewer to improve readability.

2. *The authors repeatedly use terms that are defined only in their previous publication. For example, the terms binary and bimodal in Figure 4 are defined in the authors' previous publication. These can be explained in 2-3 sentences in the current manuscript.*

Such terminology has been introduced in the revised version. Caption of Figure 4.

3. *Similarly, the authors make rather limited efforts to introduce the notion potential. What are they used for? What is the notion of "coexistence"?*

A contextualized discussion on the notion of a potential (and its possible existence or absence) has been included in the revised version. By coexistence we mean that different protein expression levels coinciding with the peaks of the bimodal distribution may coexist on a cell population. Coexistence does not comply with a deterministic description where depending on the initial condition a given protein expression steady-state is reached.

4. *The study cited to indicate that hysteresis shrinks with time is shown in "Contribution of Bistability and Noise to Cell Fate Transitions Determined by Feedback Opening" J. Mol. Biol. and the hysteresis at a single time point is shown in the cited study ([7], Cell Reports)*

We have now corrected the corresponding reference.

1 Appendix A: Phenotypic switching in gene regulatory networks (Thomas et al. 2014)

The gene regulatory network (GRN) used in [10] is depicted in Figure 1.

Figure 1: GRN in [10] to verify validity of LNA approximation for slow promoter fluctuations. Parameter values in [10] are $k_{\text{off}} = k_{\text{on}} = 0.01$, $k_{\epsilon} = k_0^{(\text{off})} = 1$, $k_m = k_0^{(\text{on})} = 10$, $\gamma_m = k_2 = 10$, $k_x = k_1 = 100$ and $\gamma_x = k_3 = 1$.

The slow promoter fluctuations described in [10] can be represented by splitting of the network in Figure 1 as in Figure 2.

Figure 2: Slow promoter fluctuations split the GRN of Figure 1 in two independent sub-networks. Parameter values are the same as in Figure 1

For the GRNs described in Figure 2, the PIDE model used by Friedman [3] admits a steady state solution which is a Gamma distribution which takes the analytic expression:

$$P(n, a, b) = \frac{n^{a-1} e^{-n/b}}{b^a \Gamma(a)}, \quad (1)$$

with $a = \frac{\text{rate of mRNA production}}{\gamma_x}$ (bursts frequency), $b = \frac{k_x}{\gamma_m}$ (bursts size) being parameters and n representing numbers of proteins. This model (with input equal to 1) is a special case of the general PIDE model introduced in [6]. This means that the production of the mRNA is constant from a given state of the promoter. For the case depicted in Figure 2 we will obtain two different values for parameter a , depending on the rate of mRNA production, which in turns depends on the promoter state (active or inactive).

In [10], the authors introduce a “cellular volume” parameter Ω , needed for the linear noise approximation that we incorporate in our model as a factor that multiplies the rate of mRNA production. Reactions in Figure 2 are written as:

where k_r is the mRNA production rate, with $k_r = \Omega k_\varepsilon$ or $k_r = \Omega k_m$. Similarly to [10], we get the stationary protein concentration $x = n/\Omega$, as a combination of the solutions for the high and low production rates, which reads:

$$p(x, \Omega) = \frac{k_{\text{off}}}{k_{\text{off}} + k_{\text{on}}} \Omega P(x\Omega, a^{\text{off}}, b) + \frac{k_{\text{on}}}{k_{\text{off}} + k_{\text{on}}} \Omega P(x\Omega, a^{\text{on}}, b), \quad (2)$$

with $a^{\text{off}} = \frac{\Omega k_\varepsilon}{\gamma_x}$ and $a^{\text{on}} = \frac{\Omega k_m}{\gamma_x}$.

In Figure 3, we compare the solutions obtained by [10] (Figure S1) with Eqn (2), for different Ω . As it can be shown in the figure, our results accurately describe the true behaviour for any range of Ω considered, whereas their approximation requires Ω to be large enough to get a reasonable accuracy.

Figure 3: First row is the Figure S1 taken from [10]. The second row depicts the solutions obtained using equation (2) for the same sets of parameters as considered in Figure S1 of [10].

References

- [1] Cai L., Friedman N. and Xie X.S. Stochastic protein expression in individual cells at the single molecule level. *Nature* 2006; 440(7082): 358-362.
- [2] Celià-Terrassa T., Bastian C., Liu D., Ell B., Aiello N. M., Wei Y., Zamalloa J., Blanco A. M., Hang X., Kunisky D., Li W., Williams E. D., Rabitz H. and Kang Y. Hysteresis control of epithelial-mesenchymal transition dynamics conveys a distinct program with enhanced metastatic ability. *Nature Communications* 2018; 9(1): 5005.
- [3] Friedman N., Cai L. and Xie X.S. Linking stochastic dynamics to population distribution: An analytical framework of gene expression. *Phys. Rev. Lett.* 2006; 97(16): 168302.
- [4] Lee J. and Lee J. Quantitative analysis of a transient dynamics of a gene regulatory network. *Phys. Rev. E* 2018; 98(6): 062404.
- [5] Lestas I., Paulsson J., Ross N.E. and Vinnicombe G. Noise in gene regulatory networks. *IEEE Trans. Autom. Control* 2008; 53(SPECIAL ISSUE): 189-200.
- [6] Pájaro M., Otero-Muras I., Alonso A.A. and Vázquez C. Stochastic modeling and numerical simulation of gene regulatory networks with protein bursting. *J. Theor. Biol.* 2017; 421: 51–70.
- [7] Pájaro M., Otero-Muras I., Vázquez C. and Alonso A.A. SELANSI: a toolbox for Simulation of Stochastic Gene Regulatory Networks. *Bioinformatics* 2018; 34(5): 893–895.
- [8] Pájaro M., Alonso A.A. and Vázquez C. Shaping protein distributions in stochastic self-regulated gene expression networks. *Phys. Rev. E* 2015; 92(3): 032712.
- [9] Scott M., Hwa T. and Ingalls B. Deterministic characterization of stochastic genetic circuits. *Proc. Natl. Acad. Sci. U.S.A.* 2007, 104(18): 7402–7407.
- [10] Thomas P., Popovic N., and Grima R. Phenotypic switching in gene regulatory networks. *Proc. Natl. Acad. Sci. U.S.A.* 2014, 111(19): 6994–6999.
- [11] To, T. - and Maheshri, N. Noise can induce bimodality in positive transcriptional feedback loops without bistability. *Science* 2010, 327(5969): 1142-1145.
- [12] Van Kampen, N.G. *Stochastic Processes in Physics and Chemistry.* 2007, Elsevier, Third Edition.
- [13] Wu M., Su R. Q., Li X., Ellis T., Lai Y. G. and Wang X. Engineering of regulated stochastic cell fate determination. *Proc. Natl. Acad. Sci. U.S.A.* 2013, 110(26): 10610-10615.

Reviewers' comments:

Reviewer #1 (Remarks to the Author):

I have read again the revised version of the paper by Pajaro et al and their response to my comments. To be clear, I believe that their calculations are technically correct and I think their results are interesting from a modeling point of view. However this paper is not about presenting a new modeling approach or its analysis rather it is trying to use a common modeling approach to reflect upon the results of certain experiments which provide evidence of hysteresis. By their own admission, their results are a proof of concept and taking everything together there is nothing in the paper and their response to convince me that the results are really applicable to real biological systems. The model they study is a toy model which provides interesting insights but which cannot be used to make wide ranging conclusions on its own, especially interpretation of experiments. Their response also shows that they do not understand clearly the linear noise approximations of which they are making statements in the paper and that they did not really look carefully at data of protein/mRNA lifetime data which is crucial to the applicability of their model. The authors are mounting a huge defense of their beloved modelling approach without really trying to think of whether the assumptions behind their master equation are biologically relevant and advocating the use of this model simply because others have already done so. I cannot hence support the publication of a paper of this type in a high impact, highly interdisciplinary journal such as Nature Communications. The paper with modifications could be suited for a more technical journal in the Nature series of journals. My more detailed comments to their responses and the revised manuscript are below:

1. The timescale separation assumption. Crucial to their Friedman PIDE model is that the mRNA is fast compared to protein. The authors mention that in a previous publication they have shown that their model remains valid under mRNA lifetimes (at least) five times shorter than protein lifetimes. The question then is whether this is constraint is consistent with experimental data. I have analysed the information from Table III of the SI of the Schwanhauser et al. Nature (2011) paper, and find that the median of the ratio of the mRNA to protein lifetime is 0.21 (using data from 4247 protein-mRNA pairs). That means that half of the protein-mRNA pairs in eukaryotic cells cannot be described by their model!!! The average of the mRNA to protein lifetime turns out to be 0.41 which again shows a general lack of clear timescale separation. Note that unlike what they have said, this decay rate data has been already corrected for protein copy numbers. In fact it is stated in the caption for Supplementary Table III: This file was replaced on 13 February 2013 - see Selbach 11848 corrigendum for details. This is also clear from reading the Corrigendum.

2. Linear-Noise Approximation (LNA) accuracy. There are comments the authors have made in their response as well as in the corrected version of the paper which are incorrect as regards to the LNA. So first of all, the LNA is not suitable only for systems near the thermodynamic limit. Although its classical derivation is based on the system-size expansion which might make one think that it is only valid in this limit, it is well known that the LNA is exact up to second-order moments for any linear system -- that means exact independent of the copy numbers of mRNA or protein. This can be easily verified by direct computation of moments from the master equation and is indeed the basis of seminal work by Paulsson and co-authors in the field of stochastic gene expression. See for example: Paulsson, Johan. "Summing up the noise in gene networks." Nature 427.6973 (2004): 415 and Lestas, Ioannis, et al. "Noise in gene regulatory networks." IEEE Transactions on Automatic Control 53.Special Issue (2008): 189-200. Besides the LNA has also been shown to be exact (up to second-order moments) for a wide range of nonlinear systems (Physical Review E 92.4 (2015): 042124) and can be extended to correct for the skewness of non-Gaussian distributions (Physical Review E 92.1 (2015): 012120). There are many developments using the system-size expansion and much of it is not found in the standard books; for a recent review I would suggest Journal of Physics A: Mathematical and Theoretical 50.9 (2017): 093001. Hence the statement that the LNA is suitable for systems near the thermodynamic limit in the paper is incorrect and should be removed. Also I would like to note that more sophisticated

methods have since appeared in the literature which give approximate time-dependence of the multivariate master equation of gene regulatory networks without invoking timescale separation between mRNA and protein or fast promoter switching (Nature communications 9.1 (2018): 3305).

3. PIDE Friedman versus conditional LNA. The authors also claim that in an Appendix to their response letter that their method does better than the conditional LNA in Thomas et al. 2014. I agree that for this particular system they chose that the conditional version of their PIDE model does slightly better in the limit of small volume than the conditional LNA. But this happens when the volume is taken so small (the case $V = 1$) that the system spends a considerable amount of its time having zero proteins (obvious by the peak at zero of the distribution). This causes in turn a small average protein number. Now this case is biologically unrealistic because we know that protein numbers are quite high in cells e.g. the data of Schwannhauser et al shows a median of 50,000 proteins. Hence their comment that the accuracy of their model is superior to the conditional LNA is incorrect since they choose parameters which are biologically irrelevant! A different study comparing the two would be needed to decide which is more accurate and I would expect that the accuracy is comparable in the vast majority of realistic conditions. Thus comments in the paper that methods based on the LNA are suitable for systems near the thermodynamic limit are incorrect and that their method is better because it is valid for purely stochastic regimes far from the thermodynamic limit are incorrect and misleading.

4. Stochastic simulation of systems with multiple genes. I completely disagree with their statement that for such systems Finite State Projection or the SSA are not efficient!! My group routinely use them and a vast repertoire of papers exists based on such methods, too many to name here but look at the extensive literature citing Journal of chemical physics 124.4 (2006): 044104 will show exactly what I mean. Also the advantage of such methods as FSP as they they can be guaranteed to converge to the correct solution with increasing state space. Hence their argument that this is why they rely on the Friedman equation instead of simulations makes no sense particularly in the light of the assumptions made by this model which as shown in point 1 above can be incompatible with common conditions in eukaryotic cells.

Conclusion:

Based on all these points, I cannot support this paper in Nature Communications. But if the authors can remove the incorrect statements about the LNA and LNA-based methods, have at least a paragraph discussing the possible limitations of their approach (the timescale separation issue in point 1 and the neglect of extrinsic noise which is common in experiments and at least as strong as intrinsic noise as papers by Michael Elowitz and others claim) as regards to interpretation of experimental data then in principle I believe that I would support its transfer to another more appropriate Nature Journal (perhaps Scientific Reports or Communications Biology).

Reviewer #2 (Remarks to the Author):

Pajaro et al improved their manuscript. Now, it become clear that they quantitatively characterize the time dependence of hysteresis with a theoretical tool that simultaneously provides a natural landscape for bistable systems based on convergence rates.

The authors have included most suggested changes but the manuscript has to be still improved for increased to clarity in order to be accessible to the broad readership of Nature Communications.

(1) The authors should display the physical units for the convergence rates and briefly discuss the practical meaning for stability (hours, days, years...).

(2) Figures that are elementary should be combined with other figures to give space to other figures. The authors could easily combine figures 1 and 2 into a single figure with two panels. At the same time, it is important to create a new figure (panel) based on the dual repressor system

(Fig S5-8). This panel should show x means and convergence rates for the dual-repressor system just as it is done for the single-activator feedback system in figure 3.

(3) The authors do not claim any more that their main discovery is the time-dependence of hysteresis, and now they correctly cite the two relevant theoretical studies (Scott et al and Lestas et al). Please note that the last author is not spelled out in the Lestas et al reference.

(4) It is very important that for each plot, the equations and the methods are clearly denoted. For example, in Fig 4 it is stated that the convergence rate is calculated by an algorithm; however, in figure 3, they calculate the eigenvalues of the matrix S_4 .

(5) All sup figure numbers should be cited in the main text (Fig. S5, S6, ...)

(6) The discussion of equations 9 and 10 can be moved to the Supp.Info.

Rebuttal Letter: Point by point responses to comments for the article entitled “**Transient hysteresis and inherent stochasticity in gene regulatory networks**”, by Manuel Pájaro, Irene Otero-Muras, Carlos Vázquez and Antonio A. Alonso.

Reviewer #1:

General Comments: *I have read again the revised version of the paper by Pajaro et al and their response to my comments. To be clear, I believe that their calculations are technically correct and I think their results are interesting from a modeling point of view. However this paper is not about presenting a new modeling approach or its analysis rather it is trying to use a common modeling approach to reflect upon the results of certain experiments which provide evidence of hysteresis. By their own admission, their results are a proof of concept and taking everything together there is nothing in the paper and their response to convince me that the results are really applicable to real biological systems. The model they study is a toy model which provides interesting insights but which cannot be used to make wide ranging conclusions on its own, especially interpretation of experiments. Their response also shows that they do not understand clearly the linear noise approximations of which they are making statements in the paper and that they did not really look carefully at data of protein/mRNA lifetime data which is crucial to the applicability of their model. The authors are mounting a huge defense of their beloved modelling approach without really trying to think of whether the assumptions behind their master equation are biologically relevant and advocating the use of this model simply because others have already done so. I cannot hence support the publication of a paper of this type in a high impact, highly interdisciplinary journal such as Nature Communications. The paper with modifications could be suited for a more technical journal in the Nature series of journals. My more detailed comments to their responses and the revised manuscript are below.*

We thank the reviewer for the comments, and the acknowledgment that our calculations are technically correct and the results interesting from a modeling point of view. The interest of our paper goes, however, beyond the modeling point of view. In order to avoid any confusion at this respect, in this second revision and response we clarify the biological relevance of our contribution.

1) The reviewer says that “by their own admission, their results are a proof of concept”. Obviously we have not been careful enough with language and it was not our intention when we used the expression “proof of concept” to diminish the biological relevance of our work. Since this expression was misleading, we have carefully rewritten the paper to avoid this type of misunderstanding.

2) The example we use in the main text to illustrate the phenomenon is not a toy example. We used a simple example (in terms of number of genes involved) for the sake of clarity. Simple doesn’t mean here non relevant from a biological point of view (a gene with autoregulation is a widely present motif in biology, and the parameters we use, taken from literature values, fall within plausible ranges). Moreover, in addition to this illustrative example we have also included a real system with parameters calibrated from experimental data where our model successfully explains what is being observed experimentally (details are presented in the SI document).

In anycase, and independently of the examples that we included in the paper, the biological

relevance of our results goes far beyond the particular examples that we show. The approach is valid for any system that fulfills the assumptions, clearly stated elsewhere. These assumptions are fulfilled for a wide range of biologically relevant systems. We discuss this in more detail below, and we included a clarifying paragraph in the new version. The reviewer, using the data in [11] says that the assumptions are not fulfilled by half of the pairs in eukaryotic cells. This is equivalent to say, using the same reasoning and data, that the assumptions are fulfilled by half of the pairs in eukaryotic cells. The assumptions are fulfilled by most cases in prokaryotic cells too. The biological relevance should be therefore out of question. We have also took the opportunity during this time to discuss with our collaborators and other colleagues working with mammalian cell lines about the copy numbers they encounter and they all agreed that in a significant number of cases the copy numbers are low (in the order of thousands or even lower). We are of course not talking about the majority of the proteins, but about a portion that is significant (see for example [6], [10], [11]). We elaborate more on this in the detailed responses.

In addition, and to avoid confusion, we clarify that we are not talking about a universal phenomenon fulfilled in any organism and under any conditions, but about what happens when a system fulfills the assumptions and copy numbers are sufficiently small. The assumptions are known and stated elsewhere and in our first version we considered that a discussion about the copy numbers of the proteins in each cell type was out of the scope of this paper. Nonetheless, and in order to address the reviewer concerns, in this new version we have introduced a paragraph discussing ranges of applicability with supporting references including [11] and others more recent. Again, these data show without room for doubt that the assumptions are fulfilled by a wide range of biologically relevant systems.

3) We do perfectly understand LNA. Actually, it is not our intention in this work to compare approaches or claim superiority of one versus another. In our view, each approach has its appropriate range and aim of application. In this particular work, for the range of applications we are interested in, our PIDE model is undoubtedly appropriate. We have justified this through precise calculations with CME and SSA. We have carefully rewritten now the text to avoid any kind of confusion at this respect.

4) “The authors are mounting a huge defense of their beloved modelling approach without trying to think of whether assumptions behind their master equation are biologically relevant and advocating the use of this model simply because others have already done so”. It is simply unfair and not objective to say that, after having read carefully our previous response. We hope to avoid this impression in this new revision, for which we have had taken the time and effort to discuss with many colleagues with extensive experience in quantitative proteomics about our results and the adequacy of our assumptions.

Detailed Comments:

1. The timescale separation assumption. Crucial to their Friedman PIDE model is that the mRNA is fast compared to protein. The authors mention that in a previous publication they have shown that their model remains valid under mRNA lifetimes (at least) five times shorter than protein lifetimes. The question then is whether this is constraint is consistent with experimental data. I have analysed the information from Table III of the SI of the Schwanhausser et al. Nature (2011) paper, and find that the median of the ratio of the mRNA to protein lifetime is 0.21 (using data from 4247 protein-mRNA pairs). That means that half of the protein-mRNA pairs in eukaryotic cells cannot be described by their model!!! The average of the mRNA to protein lifetime turns

out to be 0.41 which again shows a general lack of clear timescale separation. Note that unlike what they have said, this decay rate data has been already corrected for protein copy numbers. In fact it is stated in the caption for Supplementary Table III: This file was replaced on 13 February 2013 - see Selbach 11848 corrigendum for details. This is also clear from reading the Corrigendum.

The reviewer accepts that the assumption holds for most prokaryotic organisms (comments to the first draft). In addition, data in Schwanhausser et al (2011), gives a median ratio protein lifetime to mRNA-lifetime of 4.9 which is in the order where the model is accurate. The average ratio is even larger (8.4). Thus this assumption holds for half of the protein-mRNA pairs reported in eukaryotic cells. Therefore and contrary to the reviewer's assertion, there is a considerable amount of realistic biological systems where the assumption is fulfilled.

2. *Linear-Noise Approximation (LNA) accuracy.* There are comments the authors have made in their response as well as in the corrected version of the paper which are incorrect as regards to the LNA. So first of all, the LNA is not suitable only for systems near the thermodynamic limit. Although its classical derivation is based on the system-size expansion which might make one think that it is only valid in this limit, it is well known that the LNA is exact up to second-order moments for any linear system – that means exact independent of the copy numbers of mRNA or protein. This can be easily verified by direct computation of moments from the master equation and is indeed the basis of seminal work by Paulsson and co-authors in the field of stochastic gene expression. See for example: Paulsson, Johan. "Summing up the noise in gene networks." *Nature* 427.6973 (2004): 415 and Lestas, Ioannis, et al. "Noise in gene regulatory networks." *IEEE Transactions on Automatic Control* 53.Special Issue (2008): 189-200. Besides the LNA has also been shown to be exact (up to second-order moments) for a wide range of nonlinear systems (*Physical Review E* 92.4 (2015): 042124) and can be extended to correct for the skewness of non-Gaussian distributions (*Physical Review E* 92.1 (2015): 012120). There are many developments using the system-size expansion and much of it is not found in the standard books; for a recent review I would suggest *Journal of Physics A: Mathematical and Theoretical* 50.9 (2017): 093001. Hence the statement that the LNA is suitable for systems near the thermodynamic limit in the paper is incorrect and should be removed. Also I would like to note that more sophisticated methods have since appeared in the literature which give approximate time-dependence of the multivariate master equation of gene regulatory networks without invoking timescale separation between mRNA and protein or fast promoter switching (*Nature communications* 9.1 (2018): 3305).

The paper is not about defending one model versus another, but about understanding hysteretic behavior. The only relevant fact concerning modelling itself is that we use a model which correctly captures bimodality and we prove that with real biological systems taken from the literature (hence realistic). The discussion about LNA versus PIDE is not of relevance to the results presented. We do perfectly understand the range of validity of LNA. We completely agree with the reviewer that it is well known that the LNA is exact up to second-order moments for any linear system – that means exact independent of the copy numbers of mRNA or protein. However, our model is non-linear (as most of gene regulatory networks). Roughly speaking, stochasticity (meaning the effect of molecular noise) is important for ranges of proteins expressed in the order of thousands and lower (typical numbers for prokaryotic cells, in agreement with the literature, including works we do cite in our paper). For numbers of the order of > 10000 the systems are expected to behave as deterministic or close to the deterministic limit and intrinsic

stochasticity will not play a crucial role. However, our work is precisely on quantifying the effect of intrinsic stochasticity on hysteresis, so we are only concerned about systems in which intrinsic stochasticity is relevant (what is already clear from the title). The purpose of the paper is not about discussing when stochasticity is relevant or not, but clarify/quantifying the appearance of hysteresis in those for which the stochasticity is relevant (according to published data on protein copy numbers, this will cover most prokaryote and a significant percentage of cases in eukaryote cells). We have re-written the discussion in the paper about LNA, such that it is explicitly stated that we are referring to nonlinear systems.

PIDE Friedman versus conditional LNA. The authors also claim that in an Appendix to their response letter that their method does better than the conditional LNA in Thomas et al. 2014. I agree that for this particular system they chose that the conditional version of their PIDE model does slightly better in the limit of small volume than the conditional LNA. But this happens when the volume is taken so small (the case $V = 1$) that the system spends a considerable amount of its time having zero proteins (obvious by the peak at zero of the distribution). This causes in turn a small average protein number. Now this case is biologically unrealistic because we know that protein numbers are quite high in cells e.g. the data of Schwannhauser et al shows a median of 50,000 proteins. Hence their comment that the accuracy of their model is superior to the conditional LNA is incorrect since they choose parameters which are biologically irrelevant! A different study comparing the two would be needed to decide which is more accurate and I would expect that the accuracy is comparable in the vast majority of realistic conditions. Thus comments in the paper that methods based on the LNA are suitable for systems near the thermodynamic limit are incorrect and that their method is better because it is valid for purely stochastic regimes far from the thermodynamic limit are incorrect and misleading.

As we already stated, it was not our intention to claim the superiority of any approach versus other approach, and we have rewritten the text now to avoid this impression (see response to previous comment).

Note that we included the example discussed in [13] because the reviewer mentioned it to question PIDE validity. We have shown that the PIDE model is also valid for slow promoter transitions under the very same scenarios considered in that paper (including the case $V = 1$!).

Having said that, the PIDE model performs well for any range of proteins, and better than the LNA when the number of proteins is low. We cannot agree with the reviewer that the cases where the copy numbers are low are biologically irrelevant. In the case of prokaryotes relevance is clear (this will be in fact enough to justify the relevance of the results) but it is clear also in the case of eukaryotes. The reviewer argues that in this latter case the proteins are in high numbers in cells, and refers to the data of Schwannhauser where they show a median in copy number of 50000. With the very same data, we can see that actually a significant percentage of proteins are in low numbers. Note that 50000 is the median of the distribution (see data in the corrigendum to Schwannhauser et al 2011 doi:10.1038/nature11848, in particular the histogram with copy numbers in Fig 2b). A closer look to the histogram reveals that there is a significant percentage of proteins ($\sim 20\%$) with protein numbers lower than 5000 (see Figure 1 at the top of the next page.)

We have in fact discussed this with many colleagues and also with our collaborators at ETH and

Figure 1: Histogram with average protein copies per cell in [11] doi: 10.1038/nature11848.

PNNL working in mammalian cell signaling and using quantitative approaches to access copy numbers of proteins taking part of different pathways and in different mammalian cell lines and they agreed on the fact that in a significant number of cases copy numbers are low (in the order of thousands and lower). Therefore, we can say, very confidently, that it is a well known fact that the cases in which proteins appear at low copy numbers are a significant portion (this is actually in accordance with Schwannhäusser data, which reports $\sim 20\%$ below 5000). We are of course not talking about the majority of the proteins, but of a portion which is significant.

It is also well known that proteins in low copy numbers are sometimes absent in the surveys due to limitations in proteomics technologies. There are cases when the numbers are extremely low and then it is in fact difficult or impossible with current techniques to measure them. Note that having access to these extremely low copy numbers will shift the average and median of a potential distribution towards lower values. Also from Schwannhäusser et al [11] "Relatively few proteins had less than 100 copies per cell, indicating that some proteins of low abundance escaped detection."

In a recent study of Shi et al. [10], for example, authors report that several important proteins in the EGFR-MAPK pathway are absent in previous proteomic surveys. Quoting Shi et al. "Because of the inherent limitations of previously used proteomics technologies, both highly variable and low-abundance proteins in the EGFR-MAPK pathway frequently appeared to be absent." The authors find that two key core proteins of the pathway (SOS1 and SOS2) are only present at between 1000 and 6000 copies per cell.

Even more recently Nguyen et al [6] assert that (we quote literally): "The cellular concentrations of transcription factors (TFs) and transmembrane receptors (TMRs) are important parameters of their regulatory activity and potential. They determine cellular fate and the metabolic phenotype at multiple levels and are therefore key components in mathematical models describing the cellular decision-making. However, many so far discovered signaling pathways are composed of low abundance proteins (< 2000 copy numbers per cell, approximately 10 % of the current proteome) and only sparse information on concentrations of the involved TFs and TMRs are available. Their quantitative analysis has been a challenge in proteomics hindering analyses of signaling protein expression levels."

Stochastic simulation of systems with multiple genes. I completely disagree with their statement that for such systems Finite State Projection or the SSA are not efficient!! My group routinely use them and a vast repertoire of papers exists based on such methods, too many to name here but look at the extensive literature citing Journal of chemical physics 124.4 (2006): 044104 will show exactly what I mean. Also the advantage of such methods as FSP as they they can be guaranteed to converge to the correct solution with increasing state space. Hence their argument that this is why they rely on the Friedman equation instead of simulations makes no sense particularly in the light of the assumptions made by this model which as shown in point 1 above can be incompatible with common conditions in eukaryotic cells.

Since the goal of these paper is not comparing the efficiency of SSA versus PIDE we have removed the assertion that our approach is more efficient. The readers interested in this type of comparison can find it elsewhere or test themselves.

Having said that, doing parameter estimation, automated design of GRN or as we do in our paper, estimating the rate at which distributions converge to the stationary ones, require a large number of calls to the simulation routine (for instance to evaluate the corresponding objective function). Can that be done via brute force by FSP or Gillespie methods?. Yes, but under enormous computational effort (and computational time). Other approaches take advantage of approximations as the one we use. Again, we must remark that comparing methods of modelling and simulation in terms of efficiency is not the purpose of the paper.

Conclusion: Based on all these points, I cannot support this paper in Nature Communications. But if the authors can remove the incorrect statements about the LNA and LNA-based methods, have at least a paragraph discussing the possible limitations of their approach (the timescale separation issue in point 1 and the neglect of extrinsic noise which is common in experiments and at least as strong as intrinsic noise as papers by Michael Elowitz and others claim) as regards to interpretation of experimental data then in principle I believe that I would support its transfer to another more appropriate Nature Journal (perhaps Scientific Reports or Communications Biology).

We have now complied with all the reviewer requirements: re-written the statements about LNA and LNA-based methods, we have added a paragraph with the time scale separation assumption (bursting assumption was already mentioned, but now we state it in numerical terms also in the last paragraph) and we explicitly stated that we are not considering extrinsic noise. Importantly, note that we do not consider other sources of noise because our goal is quantifying the effect of the intrinsic noise in hysteresis. Based on our responses to the previous questions (which have been also reflected in the new version of the paper), the biological relevance of the results is clearly justified, and we do not agree with transferring our work to other journal.

Reviewer #2:

Pajaro et al improved their manuscript. Now, it become clear that they quantitatively characterize the time dependence of hysteresis with a theoretical tool that simultaneously provides a natural landscape

for bistable systems based on convergence rates. The authors have included most suggested changes but the manuscript has to be still improved for increased clarity in order to be accessible to the broad readership of Nature Communications.

We thank the reviewer for the helpful comments that allowed us to improve the manuscript. In this new version we hope to have improved the manuscript as requested in terms of clarity and accessibility.

(1) The authors should display the physical units for the convergence rates and briefly discuss the practical meaning for stability (hours, days, years. . .).

Convergence rates are expressed in units of inverse of time. This has been explicitly indicated in page 8 after equation (8), Figure 2 and Figure 5.

(2) Figures that are elementary should be combined with other figures to give space to other figures. The authors could easily combine figures 1 and 2 into a single figure with two panels. At the same time, it is important to create a new figure (panel) based on the dual repressor system (Fig S5-8). This panel should show x means and convergence rates for the dual-repressor system just as it is done for the single-activator feedback system in figure 3.

As suggested by the reviewer, Figures 1 and 2 have been combined. In the corrected version Figure 1 and 2 are Figure 1 A and 1 B respectively.

The dual repressor system has been moved to the main text with previous Figure S5 being now Figure 4. A substantial part of the discussion, previously in section “Mutual inhibitory gene regulatory motif in yeast” of the SI has been moved to the main text (see also comment 5 below). The updated version includes a new figure (Figure 5) showing the convergence rate η (equation (8)) for the 2D system. Note that as it happens for the 1D example, the convergence rates are smaller in the parameter region where bimodal distributions appear.

(3) The authors do not claim any more that their main discovery is the time-dependence of hysteresis, and now they correctly cite the two relevant theoretical studies (Scott et al and Lestas et al). Please note that the last author is not spelled out in the Lestas et al reference.

In the new version we have corrected the reference.

(4) It is very important that for each plot, the equations and the methods are clearly denoted. For example, in Fig 4 it is stated that the convergence rate is calculated by an algorithm; however, in figure 3, they calculate the eigenvalues of the matrix S_4 .

As recommended by the reviewer the methods employed are explicitly indicated in the corresponding figure captions.

(5) All sup figure numbers should be cited in the main text (Fig. S5, S6, ...)

We have included the discussion of all supplementary figures in the main text.

(6) The discussion of equations 9 and 10 can be moved to the Supp.Info

We have moved the discussion to the supplementary material as a Supplementary Note 2 entitled “Correspondence between deterministic and stochastic counterparts”.

References

- [1] Cai L., Friedman N. and Xie X.S. Stochastic protein expression in individual cells at the single molecule level. *Nature* 2006; 440(7082): 358-362.
- [2] Celià-Terrassa T., Bastian C., Liu D., Ell B., Aiello N. M., Wei Y., Zamalloa J., Blanco A. M., Hang X., Kunisky D., Li W., Williams E. D., Rabitz H. and Kang Y. Hysteresis control of epithelial-mesenchymal transition dynamics conveys a distinct program with enhanced metastatic ability. *Nature Communications* 2018; 9(1): 5005.
- [3] Friedman N., Cai L. and Xie X.S. Linking stochastic dynamics to population distribution: An analytical framework of gene expression. *Phys. Rev. Lett.* 2006; 97(16): 168302.
- [4] Lee J. and Lee J. Quantitative analysis of a transient dynamics of a gene regulatory network. *Phys. Rev. E* 2018; 98(6): 062404.
- [5] Lestas I., Paulsson J., Ross N.E. and Vinnicombe G. Noise in gene regulatory networks. *IEEE Trans. Autom. Control* 2008; 53(SPECIAL ISSUE): 189-200.
- [6] Nguyen C.D.L., Malchow S., Reich S., Steltgens S., Shuvaev K.V., Loroach S., Lorenz S., Sickmann A., Knobbe-Thomsen C.B., Tews B., Medenbach J, and Ahrends R. A sensitive and simple targeted proteomics approach to quantify transcription factor and membrane proteins of the unfolded protein response pathway in glioblastoma cells *Sci Rep* 2019; 9:8836.
- [7] Pájaro M., Otero-Muras I., Alonso A.A. and Vázquez C. Stochastic modeling and numerical simulation of gene regulatory networks with protein bursting. *J. Theor. Biol.* 2017; 421: 51–70.
- [8] Pájaro M., Otero-Muras I., Vázquez C. and Alonso A.A. SELANSI: a toolbox for Simulation of Stochastic Gene Regulatory Networks. *Bioinformatics* 2018; 34(5): 893–895.
- [9] Pájaro M., Alonso A.A. and Vázquez C. Shaping protein distributions in stochastic self-regulated gene expression networks. *Phys. Rev. E* 2015; 92(3): 032712.
- [10] Shi T., Niepel M., McDermott J.E., Gao Y., Nicora C.D., Chrisler W.B., Markillie L.M., Petyuk V.A., Smith R.D., Rodland K.D., Sorger P.K., Qian W.J and Wiley S. Conservation of protein abundance patterns reveals the regulatory architecture of the EGFR-MAPK pathway. *Science Signaling* 2016; 9(436): rs6.
- [11] Schwanhäusser B., Busse D., Li N., Dittmar G., Schuchhardt J., Wolf J., Chen W. and Selbach M. Global quantification of mammalian gene expression control. *Nature* 2011; 473(7347): 337-342.
- [12] Scott M., Hwa T. and Ingalls B. Deterministic characterization of stochastic genetic circuits. *Proc. Natl. Acad. Sci. U.S.A.* 2007, 104(18): 7402–7407.
- [13] Thomas P., Popovic N., and Grima R. Phenotypic switching in gene regulatory networks. *Proc. Natl. Acad. Sci. U.S.A.* 2014, 111(19): 6994–6999.
- [14] To, T. -. and Maheshri, N. Noise can induce bimodality in positive transcriptional feedback loops without bistability. *Science* 2010, 327(5969): 1142-1145.
- [15] Van Kampen, N.G. Stochastic Processes in Physics and Chemistry. 2007, Elsevier, Third Edition.

- [16] Wu M., Su R. Q., Li X., Ellis T., Lai Y. G. and Wang X. Engineering of regulated stochastic cell fate determination. *Proc. Natl. Acad. Sci. U.S.A.* 2013, 110(26): 10610-10615.